# Coriolic Acid (13-(*S*)-Hydroxy-9*Z*, 11*E*-octadecadienoic Acid) from Glasswort (*Salicornia herbacea* L.) Suppresses Breast Cancer Stem Cell through the Regulation of c-Myc

**DOI:** 10.3390/molecules25214950

**Published:** 2020-10-26

**Authors:** Yu-Chan Ko, Hack Sun Choi, Ji-Hyang Kim, Su-Lim Kim, Bong-Sik Yun, Dong-Sun Lee

**Affiliations:** 1Interdisciplinary Graduate Program in Advanced Convergence Technology and Science, Jeju National University, Jeju 63243, Korea; koyuchan94@gmail.com (Y.-C.K.); seogwi12@naver.com (J.-H.K.); ksl1101@naver.com (S.-L.K.); 2Subtropical/Tropical Organism Gene Bank, Jeju National University, Jeju 63243, Korea; choix074@jejunu.ac.kr; 3Division of Biotechnology, College of Environmental and Bioresource Sciences, Chonbuk National University, Gobong-ro 79, Iksan 54596, Korea; bsyun@jbnu.ac.kr; 4Practical Translational Research Center, Jeju National University, Jeju 63243, Korea; 5Faculty of Biotechnology, College of Applied Life Sciences, Jeju National University, SARI, Jeju 63243, Korea

**Keywords:** breast cancer stem cells (BCSCs), coriolic acid, c-Myc, mammospheres

## Abstract

Cancer stem cells have certain characteristics, such as self-renewal, differentiation, and drug resistance, which are related to tumor progression, maintenance, recurrence, and metastasis. In our study, we targeted breast cancer stem cells (BCSCs) using a natural compound, coriolic acid, from *Salicornia herbacea* L. This compound was isolated by mammosphere formation inhibition bioassay-guided fractionation and identified by using NMR spectroscopy and electrospray ionization mass spectrometry. Coriolic acid inhibited the formation of mammospheres and induced BCSC apoptosis. It also decreased the subpopulation of CD44^high^/CD24^low^ cells, a cancer stem cell (CSC) phenotype, and specific genes related to CSCs, such as *Nanog,*
*Oct4*, and *CD44*. Coriolic acid decreased the transcriptional and translational levels of the c-Myc gene, which is a CSC survival factor. These results indicated that coriolic acid could be a novel compound to target BCSCs via regulation of c-Myc.

## 1. Introduction

Triple-negative breast cancer (TNBC) cells are cancer cells that lack estrogen receptor (ER), progesterone receptor (PR), and human epidermal growth factor receptor 2 (HER2) expression [1]. Only 10~15% of breast cancers are TNBC [2]. Additionally, cancer stem cells or tumor stem cells have the properties of self-renewal and differentiation [3]. The existence of cancer stem cells in a subpopulation of cancer cells plays a key role in the disruption of cancer therapy [4]. Therefore, many studies have aimed to target cancer stem cells [5]. In particular, breast cancer stem cells (BCSCs) express the distinct markers aldehyde dehydrogenase 1, CD44, and CD24 [6,7].

c-Myc, a proto-oncogene, is a regulatory gene that binds to enhancer box sequences (E-boxes) through dimerization with Myc-associated factor X (MAX) [8,9]. Transcription of c-Myc genes is initiated at different sites. Depending on the site, one of three major c-Myc proteins can be transcribed: c-Myc1, c-Myc2, or c-MycS [10]. c-Myc is related to cell proliferation, differentiation, apoptosis, and metabolism [11]. Downregulation of c-Myc results in apoptosis of glioma cancer stem cells [12] and inhibition of breast cancer stem cell formation [13]. Thus, an effective therapeutic strategy against cancer stem cells (CSCs) could be through the study of the inhibition of c-Myc [14].

*Salicornia herbacea* L. is a salt tolerant plant that grows in salt fields or reclaimed areas around the coasts of Korea, China, and the United States. This plant has a rich chemical composition; in 100 g of dry sample, the moisture content is 90.9% and there is 1888.8 mg of sodium, 650 mg of calcium, 650 mg of potassium, 84.8 mg of iron, 70 mg of iodine, 50 mg of magnesium, and 29.6 mg of zinc [15]. Additionally, *S. herbacea* L. contains saponins and flavonoids, which are useful for humans [16,17,18,19]. Moreover, *S. herbacea* L. has anti-inflammatory, anticancer, and antioxidant activities [20,21,22]. However, there are no studies that have researched the cancer stem cell-targeting mechanism of *S. herbacea* L.

In this study, *S. herbacea* L. was used to target BCSCs. We purified coriolic acid from the plant and examined its anti-CSC activity. Coriolic acid inhibited both the regulation of c-Myc at the transcriptional level and protein level and the proliferation of MDA-MB-231 BCSCs.

## 2. Results

### 2.1. A CSC Inhibitor was Isolated from S. herbacea L.

The compound from *S. herbacea* L. that inhibited BCSCs was isolated by CSC assay-guided purification as shown in Figure 1A. We followed the isolation protocol using methanol/ethyl acetate extraction, silica gel chromatography, Sephadex gel filtration, and thin-layer chromatography (TLC). The sample was finally isolated as a single compound (Figure 1B) and showed inhibition of mammosphere formation (Figure 1C). The compound was identified as coriolic acid (Figure 2).

### 2.2. Coriolic Acid Suppresses the Growth of MDA-MB-231 and MCF-7 Cells and the Formation of Mammospheres

Breast cancer cell lines (MDA-MB-231 and MCF-7) were incubated with increasing concentrations of coriolic acid for 24 h. The antiproliferative effects of coriolic acid were examined by (3-(4,5-dimethylthiazol-2-yl)-5-(3-carboxymethoxyphenyl)-2-(4-sulfophenyl)-2H-tetrazolium) (MTS) assay. Coriolic acid inhibited the proliferation of MDA-MB-231 and MCF-7 cells in a dose-dependent manner. The doses of coriolic acid causing 50% growth inhibition (IC_50_) of MDA-MB-231 and MCF-7 cells at 24 h incubations were 289.3 and 386.9 µM. Coriolic acid exerted fewer cytotoxic effects in human embryonic kidney cells, human embryonic kidney 293 cells (HEK 293 cells) as coriolic acid did not affect the proliferation of HEK-293 cells, as shown in Figure 3A. Coriolic acid also inhibited colony formation and migration of breast cancer cells (Figure 3B,C). To examine whether coriolic acid can inhibit the formation of mammospheres, primary BCSCs were treated with this compound, as shown in Figure 3D, and coriolic acid reduced both the number of mammospheres as well as their size.

### 2.3. Coriolic Acid Reduces the CD44^high^/CD24^low^-Expressing Population

Because the CD44^high^/CD24^low^ subpopulation level is a marker of breast CSCs, we examined the MDA-MB-231 BCSC marker CD44^high^/CD24^low^ subpopulation. We treated MDA-MB-231 cells with 200 µM coriolic acid. Coriolic acid decreased the subpopulation of the CD44^high^/CD24^low^-expressing MDA-MB-231 cells from 94.0% to 66.3% (Figure 4). Coriolic acid selectively killed cancer stem cell of heterogeneous cancer cells. This result revealed that coriolic acid modestly reduced the frequency of a CSC trait.

### 2.4. Coriolic Acid Induces Apoptosis in BCSCs and Suppresses CSC Marker Gene Expression and Mammosphere Growth

To evaluate the effects of coriolic acid on apoptosis in mammospheres, we treated MDA-MB-231 mammospheres with 150 µM coriolic acid. The subpopulation of cells in early apoptosis increased from 1.8% to 5.1%, and the subpopulation of cells in late apoptosis also increased from 5.7% to 39.6%, as shown in Figure 5A. Additionally, the expression of CSC marker genes was inhibited by coriolic acid. Coriolic acid-treated samples reduced the expression of Nanog, CD44, c-myc, and Oct4 in mammospheres (Figure 5B). To examine whether coriolic acid inhibits mammosphere proliferation, we treated BCSCs with coriolic acid for 2 days. After counting the number of cells, we cultured the same number of BCSCs. As shown in Figure 5C, coriolic acid led to the inhibition of mammosphere proliferation.

### 2.5. Effects of Coriolic Acid on the Regulation of c-Myc

c-Myc has been studied for new therapeutic opportunities against cancer [23]. Therefore, the effects of coriolic acid on c-myc gene expression were examined in BCSCs. Treatment with coriolic acid reduced the c-myc transcripts in MDA-MB-231 mammospheres (Figure 6A). Coriolic acid also downregulated the total protein expression level of c-Myc (Figure 6B). We extracted the nuclear fraction from BCSCs and measured c-Myc protein expression. The c-Myc protein level in both the cytosolic fraction and the nuclear fraction significantly decreased, as shown in Figure 6C. To examine the effects of c-Myc as a survival factor in MDA-MB-231 BCSCs, we used c-Myc-specific siRNA. Breast cancer cells were transfected with siRNA to silence c-Myc, which led to a decline in the MFE as shown in Figure 6D. Both the size (<60 μm) and the number of MDA-MB-231 BCSCs were significantly reduced. In conclusion, c-Myc is a key protein involved in mammosphere formation, and coriolic acid regulates c-Myc in MDA-MB-231 BCSCs (Figure 7).

## 3. Discussion

*S. herbacea* L. is a salt-tolerant glasswort species from the southwest coast of Korea, China, and the United States. It contains salt and many minerals and is taken as a traditional medicine for obesity, diabetes, and gastroenteric disorders in Korea [24]. Several studies have investigated the biological effects of this plant, including its antioxidant, anti-inflammatory, and anticancer effects [25,26]. First, we extracted the *S. herbacea* L. powder by bioassay-guided fractionation and determined that the CSC inhibitor from this plant is coriolic acid. Coriolic acid is a hydroxyl fatty acid also known as (9*Z*,11*E*)-13-hydroxy-9,11-octadecadienoic acid, or 13-HODE. It can inhibit the metastasis of cancer cells by preventing their adhesion to endothelial cells [27]. The arachidonic acid (AA) pathway and linoleic acid (LA) pathway have been considered as important contributors to colon cancer growth. 13-HODE (coriolic acid) inhibited both murine and human colon cancer cell proliferation. Human 15-lipoxygenase 1 (15-LOX-1) converts LA to anti-tumor 13-*S*-hydroxyoctadecadienoic acid (13-HODE) and 15-LOX-2 converts AA to 15-hydroxyeicosatetraenoic acid (15-HETE). The 15-HETE has no antitumor activity [28]. Conjugated linoleic acid (CLA) may be regarded as a component of the diet that exerts anti-proliferative or pro-apoptotic activity [29]. Additionally, coriolic acid is related to granulocyte colony stimulating factor and interleukin-6 (IL-6) [30]. However, there are no literature reports on coriolic acid and BCSCs. This is the first study to show that *S. herbacea* L. inhibits BCSCs.

TNBC is an aggressive subtype of cancer that lacks the ER, the PR, and the HER2 receptor [31]. Young women are easily affected by TNBC. The rates of relapse are higher in TNBC and the overall survival times are shorter than those of other breast cancer subtypes [32]. BCSCs are a subpopulation of breast cancer cells with properties such as self-renewal and differentiation [33]. BCSCs have been identified as having CD44^high^/CD24^low^ expression and aldehyde dehydrogenase (ALDH) activity [34]. Because BCSCs are related to poor prognosis, studying the mechanism of BCSCs is urgently needed [35]. We examined in which cells coriolic acid from *S. herbacea* L. inhibits BCSC formation, and we found that the proliferation of the breast cancer cell lines MDA-MB-231 and MCF-7 was inhibited by treatment with coriolic acid. The efficiency of BCSC formation was also inhibited in coriolic acid-treated cells (Figure 3). Moreover, the BCSC marker CD44^high^/CD24^low^ subpopulation was reduced in coriolic acid-treated cells (Figure 4). These data showed that coriolic acid could be an anticancer agent and therapy. The lipid extracts containing α-linoleic acid (64.6%), α-linolenic acid (14.6%), and oleic acid (12.6%) have properties as an anti-cancer stem cell regulator via suppression of the self-renewal capacity [36].

The transcription factor c-Myc is related to cell growth, differentiation, and apoptosis through the regulation of many target genes [37]. Myc family proteins contain c-Myc, n-Myc, and l-Myc. Among them, c-Myc can become a promising therapeutic target molecule in cancer. c-Myc mediates drug resistance of colorectal CSCs [14]. Degradation of the c-Myc protein results from ubiquitin-dependent or ubiquitin-independent pathways, which leads to short-lived proteins [38,39]. Interestingly, the correlation between the transcriptional level of c-myc and the occurrence of apoptosis in hepatocellular carcinoma (HCC) cells was researched [40]. Triptolide (C1572), a small-molecule natural compound, selectively kills CSCs in human triple-negative breast cancer (TNBC) cell lines. Nanomolar concentrations of C1572 dramatically reduced c-MYC (MYC) protein levels via a proteasome-dependent mechanism. C1572 as a promising therapeutic agent eradicate CSCs for drug-resistant TNBC treatment [41]. In TNBC stem cells, the downregulation of MYC induced CSC depletion [41]. In this study, coriolic acid decreased the transcriptional level of c-myc and induced the reduction of the c-Myc protein in BCSCs (Figure 6) in ubiquitin-independent pathway. Coriolic acid also inhibited the proliferation of BCSCs and the gene expression of CSC markers (Figure 5).

Many fatty acids are contained within the human body and our diet. They have biological effects, such as those on cellular function, metabolism, and responsiveness [42]. A number of anticancer drugs regarded as lipid-based molecules are the most promising group of agents [43]. Conjugated linoleic acid has an inhibitory effect on cell proliferation and murine mammary tumorigenesis [44,45,46]. Long-chain N-3 fatty acids are related to breast cancer inhibition through the activation of peroxisome proliferator-activated receptor-γ (PPAR-γ), which causes apoptosis [47]. Additionally, omega-3 fatty acids have an effect on the prevention of breast cancer [48]. Linoleic acid inhibits growth of colorectal cancer cell by oxidative stress and mitochondrial dysfunction [49]. Arachidonic and eicosapentaenoic acids induce oxidative stress to inhibit the growth of human glioma cells [50]. Thus, fatty acids could be effective for decreasing the risk of accumulating tumor-initiating cells [51]. Unsaturated fatty acids showed cytotoxicity against a SNU16 human stomach cancer cell line and inhibits transcription factor (Myc–Max dimer)–DNA complex formation. Conjugated linoleic acid suppressed mRNA expression of several myc-target genes; ornithine decarboxylase, p53, cdc25a in the SNU16 cells [52]. Celastrol, triterpenoids, inhibits interaction of Myc-Max heterodimers and its DNA binding. Celastrol and SBI compounds inhibit cell proliferation, promote depletion of Myc protein and selectively inhibit a Myc-responsive promoter [53]. We need further study whether coriolic acid inhibits interaction of Myc-Max heterodimers, its DNA binding, and a Myc-responsive promoter. These studies have shown that fatty acids can be useful for cancer and cancer stem cell therapy. Our study suggests that targeting c-Myc may play a key role in breast cancer therapy and that coriolic acid inhibited BCSC formation by regulating c-Myc.

## 4. Materials and Methods

### 4.1. Reagents

Silica gel 60A (Analtech, Newark, DE, USA) and silica thin-layer chromatography (TLC) plates were purchased from Merck KGaA (Darmstadt, Germany), and Sephadex LH-20 resin was purchased from Sigma (St. Louis, MO, USA). High-performance liquid chromatography (HPLC) was performed on a Shimadzu 20A system (Shimadzu, Kyoto, Japan). Cell growth was assayed using the CellTiter 96^®^ Aqueous Solution cell proliferation assay kit (Promega, WI, USA). We obtained coriolic acid by purification of *S. herbacea* L. extracts.

### 4.2. Plant Material

*S. herbacea L.* (glasswort) was obtained from Dasarang Ltd. (Sinan, Korea) and was freeze-dried and ground. A glasswort sample (no. 2019_15) was deposited at the Jeju Center of Korea Basic Science Institute, Jeju National University (KBSI, Core-facility center, Jeju, Korea).

### 4.3. Isolation of Coriolic Acid

*S. herbacea* L. (2 kg) was extracted with 100% methanol at 30 °C overnight. The purification procedure was bioassay-based and is summarized in Figure 1A. After the methanol extracts were mixed with distilled water, the methanol was evaporated. The water-suspended extracts were mixed and extracted with the same volume of ethyl acetate. After evaporation of the ethyl acetate, the sample was dissolved in methanol, loaded onto a silica gel column (3 × 35 cm) and fractionated with a solvent mixture (chloroform:methanol 30:1) (Appendix A). The sample was separated into six parts, dissolved in methanol and assayed for potential activity against mammosphere formation. Part #6 suppressed MDA-MB-231 cell mammosphere formation. Part #6 was therefore loaded onto a Sephadex LH-20 gel filtration column (2.5 × 30 cm) and divided into four fractions (Appendix A). Each fraction was tested to evaluate its ability to inhibit mammosphere formation, and fraction #2 showed mammosphere formation inhibitory activity. Fraction #2 was subjected to preparatory TLC (glass plate; 20 × 20 cm) in a glass TLC chamber. Each band was isolated from the TLC plate and assayed by examining mammosphere formation (Appendix A). Fraction #2 was then loaded onto a preparatory HPLC LC-20A (Shimadzu, Tokyo, Japan) with an octadecyl-silica (ODS) 10 × 250 mm C18 column (Shim-pack GIS C-18; pore size; 100 Å, particle size; 10μm and flow rate; 3 mL/min). For elution, the acetonitrile proportion of the mobile phase was initially set to 20%, then increased to 60% over 20 min, and finally increased to 100% over 30 min (Appendix A). The isolated sample was detected at a retention time of 31.5 min (Figure 1B).

### 4.4. Structural Analysis of the Purified Sample

The chemical structure of the isolated compound was determined by mass spectrometry and NMR measurements. The molecular weight was 296 by electrospray ionization (ESI) mass spectrometry, which showed a quasi-molecular ion peak at m/z 295.4 [M − H]^-^ in negative mode (see Appendix A). The ^1^H NMR spectrum in CD_3_OD showed signals from four olefinic methines at δ 6.50, 5.98, 5.62, and 5.41 ppm, one oxygenated methine at δ 4.08 ppm, 11 methylenes at δ 2.27, 2.20, 1.60, 1.40, 1.54/1.48, and 1.25–1.40 ppm, and one methyl at δ 0.91 ppm. In the ^13^C NMR spectrum, the 18 carbon peaks included one carbonyl carbon at δ 178.1 ppm, four olefinic methine carbons at δ 137.5, 133.0, 129.5, and 126.7 ppm, one oxygenated methine carbon at δ 73.5 ppm, 11 methylene carbons at δ 38.6, 35.2, 33.1, 30.8, 30.4, 30.4, 40.4, 28.7, 26.4, 26.3, and 23.8 ppm, and one methyl carbon at δ 14.6 ppm (see Appendix A). All proton-bearing carbons were assigned by the HMQC spectrum, and the ^1^H-^1^H COSY spectrum revealed three partial structures, CH_3_-CH_2_-, -CH_2_-CH_2_-CH(-O)-CH=CH-CH=CH-CH_2_-CH_2_-, and -CH_2_-CH_2_-CH_2_- (see Appendix A). Further structural elucidation was performed with the aid of the HMBC spectrum, which showed long-range correlations from the methyl protons at δ 0.91 ppm to the carbons at δ 33.1 and 23.8 ppm and from the methylene protons at δ 1.54/1.48 ppm to the carbons at δ 33.1 and 26.4 ppm. Finally, the methylene protons at δ 2.27 and 1.60 ppm showed a long-range correlation to the carbon at δ 178.1 ppm (see Appendix A). The geometries of C-9 and C-11 were established as cis and trans by the proton coupling constants of 11.0 and 15.5 Hz, respectively. Therefore, the isolated compound was identified as coriolic acid (Figure 2, Appendix A). 

### 4.5. Breast Cancer Cell and Mammosphere Cultures

MDA-MB-231, MCF-7, and HEK-293 cells were obtained from the American Type Culture Collection (Rockville, MD, USA) and cultured in Dulbecco’s modified Eagle medium (DMEM) supplemented with 10% fetal bovine serum (Gibco Thermo Fisher Scientific, CA, USA) and 1% penicillin/streptomycin (Gibco, Thermo Fisher Scientific, CA, USA). Cancer cells (1 × 10^4^ or 4 × 10^4^/well) were cultured in an ultralow-attachment 6-well plate with MammoCult™ culture medium (STEMCELL Technologies, Vancouver, BC, Canada). All cancer cells were cultured in a humidified 5% CO_2_ incubator at 37 °C. The formation of mammospheres was assessed by the NIST’s integrated colony enumerator (NICE) software program and examined by assessing the mammosphere formation efficiency (MFE) (% of control) [54].

### 4.6. Cell Viability Assay

MDA-MB-231, MCF-7, and HEK-293 cells were seeded in a 96-well plate for 24 h. The breast cancer cell lines were treated with increasing concentrations (0, 20, 40, 80, 100, 150, 200, 300, and 400 µM) coriolic acid, and HEK-293 cells were treated with various concentrations (0, 100, 200, 300, 400 µM) coriolic acid for 24 h in cell culture medium. Then, we followed the cell viability assay manufacturer’s protocol [55]. CellTiter 96^®^TM Aqueous One Solution (Promega, Madison, WI, USA) was used for cell viability assay. After mixing DMEM and aqueous one solution (5:1), we added 100 µL of mixture to each well and incubated the cells at 37◦C for 1h. The OD_490_ was measured using a Versa Max ELISA microplate reader (Molecular Devices, San Jose, CA, USA).

### 4.7. Colony Formation Assay

Breast cancer cells were cultured in a 6-well plate at with 2 × 10^3^/well and treated with 200 µM coriolic acid. The cells were cultured for 1 week at 37 °C. The colonies were washed three times with 1X phosphate-buffered saline (PBS), fixed for 10 min using 4% formaldehyde, and stained for 1 h with 0.04% crystal violet. After washing twice with distilled water, we acquired images using a scanner (Epson Perfection, Epson, Tokyo, Japan). The colonies were counted with the NICE software program [56].

### 4.8. Migration Assay

MDA-MB-231 and MCF-7 cells were cultured in a 6-well plate with 2 × 10^6^ cells/plate. After 24 h, the cells were scratched using a microtip. The cells were washed two times with 1X PBS and cultured with coriolic acid in DMEM for 16 h. Photographs of the migrated areas were acquired using a light microscope.

### 4.9. Flow Cytometry Analysis

After treatment with coriolic acid for 24 h, cancer cells were detached by using 1X trypsin/EDTA following a previously described method [57]. The detached cells were washed with activated cell sorting (FACS) buffer and suspended with 100 µL of FACS buffer. We added 10 µL of FITC-conjugated anti-human CD44 and phycoerythrin (PE)-conjugated anti-human CD24 to each samples. The samples were incubated on ice for 20 min, washed two times with 1X FACS buffer, and the isolated cells were then centrifuged and washed two times with 1X FACS buffer. The cell pellet was analyzed using an Accuri C6 flow cytometer (BD, San Jose, CA, USA).

### 4.10. Gene Expression

Total RNA was isolated using MDA-MB-231 cells. RT-qPCR was performed using a real-time One-Step RT-qPCR kit (Enzynomics, Daejeon, Korea). We followed a previously described method, and we made RT-qPCR mixture containing TOPrealTM One-step RT qPCR Enzyme MIX 1 µL, 2X TOPrealTM One-step RT qPCR Reaction MIX (with low ROX) 10 µL, RNA template (100 ng/µL) 1 µL, specific primers-F (10 ng/µL) 1 µL. specific primers-R (10 ng/µL) 1 µL, and sterile water 6 µL in each samples. The relative transcript expression levels of the target genes were analyzed using the comparative CT method [57]. The list of the specific primers is shown in Appendix A. PCR experiments were tested to allow statistical analysis. The β-actin gene was used as an internal control.

### 4.11. Immunoblot Analysis

Protein samples with/without coriolic acid were extracted from breast mammospheres and cancer cells. We cultured MDA-MB-231 cancer stem cells to ultra-low attachment 6 well plate for 5 days and treated 150 µM of coriolic acid for 2 days. We washed the cell pellet twice for 1X PBS. The cells were resuspended in buffer A (pH 7.9 of 10mM HEPES, 1.5 mM MgCl_2_, 10 mM KCl, 0.05% NP-40, 0.5 mM DTT, 10 mM protease inhibitor, 10 mM NaF, and 10 mM Vanadate) and the lysate was microcentrifuged at 10,000× *g* for 5 min to pellet the nuclei. The supernatant contains cytosol fraction and the resulting pellet contains the nucleus. The nuclear pellet was dissolved with RIPA buffer with 10 mM protease inhibitor, 10 mM NaF, and 10 mM Vanadate and then pipetting. The samples were put on ice for 30 min and then microcentrifuged at 14,000× *g* for 15 min. The resulting supernatant contains the nuclei. The proteins were separated by using a 10% SDS-PAGE gel and SDS-PAGE was conducted using tris-glycine buffer. After transferred, the polyvinylidene fluoride (PVDF) membranes (Millipore, Burlington, MA, USA) were incubated with Odyssey blocking buffer at room temperature for 1h and then incubated overnight with primary antibodies at 4 °C. The primary antibodies were diluted one thousand to one with primary antibody diluent (mixture of Odyssey blocking buffer and 0.2% tween-20). The primary antibodies were anti-c-Myc, anti-lamin B (Cell Signaling Technology, Danvers, MA, USA), and anti-β-actin (Santa Cruz Biotechnology, Dallas, TX, USA). After the PVDF membranes were washed three times using PBS-Tween 20 (0.1%, *v*/*v*), all membranes were incubated with IRDye 680RD- and IRDye 800W-labeled secondary antibodies with secondary antibody diluent (mixture of Odyssey blocking buffer, 0.2% tween-20, and 0.01% SDS) for 1 h at room temperature. The signals were detected with an Odyssey CLx imaging system (LI-COR, Lincoln, NE, USA).

### 4.12. SiRNA of c-Myc

MDA-MB-231 cells were cultured in a 6-well plate at a density of 1.0 × 10^6^ cells/plate. To examine the role of c-Myc in mammosphere formation, MDA-MB-231 cells were transfected with siRNAs targeting the human c-Myc gene (Bioneer, Daejeon, Korea). The c-Myc siRNAs (NM_002467.3) and a scrambled siRNA were obtained from Bioneer (Daejeon Cor., Korea). For siRNA transfection, MDA-MB-231 cells were transfected using Lipofectamine 3000 (Thermo Scientific, Waltham, MA, USA), according to the manufacturer’s instructions. For siRNA transfection, we cultured MDA-MB-231 cells to be 70~90% confluent. After the cells were attached to plate, we diluted Lipofectamine^®^ 3000 reagent 4 µL in Opti- Opti-Minimal Essential Medium (MEM)^®^ medium 125 µL and prepared master mix of SiRNA by diluting SiRNA 5µg in Opti-MEM^®^ medium 125 µL in each tube. Then, we mixed the diluted SiRNA and diluted Lipofectamine^®^ 3000 reagent (for control, only Opti-MEM^®^ and diluted Lipofectamine^®^ 3000 should be mixed with a scrambled siRNA) and incubated it for 5 min at room temperature. We added SiRNA-lipid complex to each wells and incubated cells for 2~4 days at 37 °C. Then, analyze transfected cells. The protein level of c-Myc was detected with the c-Myc antibody.

### 4.13. Statistical Analysis

All data from three independent experiments are reported as the mean ± SD. One-way ANOVA was used for statistical analysis. The data were analyzed with GraphPad Prism 5.0 software (GraphPad software, Inc., San Diego, CA, USA).

## 5. Conclusions

A BCSC-inhibiting compound from *S. herbacea* extracts was purified using silica gel, gel filtration, TLC, and HPLC. The compound isolated from glasswort extracts was identified as coriolic acid, a mammosphere formation inhibitor using by mass spectrometry and NMR spectroscopy. Coriolic acid inhibits the proliferation, migration, colony formation of breast cancer cells, formation of mammosphere, breast cancer stem cell formation, the CD44^high^/CD24^low^ subpopulation, and the transcriptional levels of CSC-related genes. This compound regulates c-Myc, which is a proto-oncogene and a survival factor of CSCs. Our data suggested that coriolic acid suppresses c-Myc function and may be an inhibitory compound against BCSCs. Coriolic acid could, thus, be a novel tool for breast cancer therapy

## Figures and Tables

**Figure 1 molecules-25-04950-f001:**
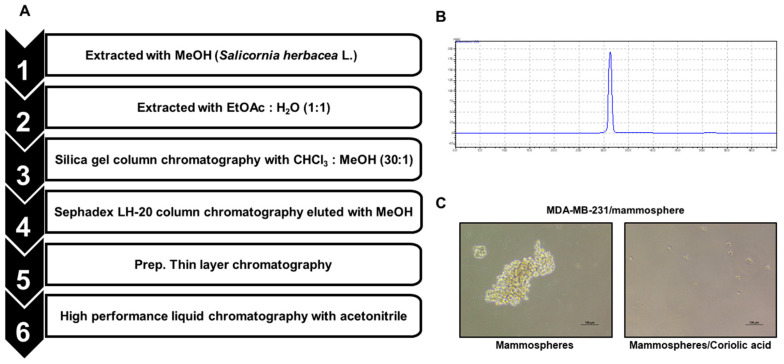
Procedures for the isolation of the breast cancer stem cell (BCSC) inhibitor isolated from *S. herbacea* L. and the mammosphere formation assay with the purified sample. (**A**) Purification flowchart of the BCSC inhibitor. (**B**) HPLC analysis of BCSC inhibitor derived from *S. herbacea* L. (**C**) Mammosphere formation assay using the isolated sample. Photos show exemplary mammospheres and were captured by inverted light microscopy (scale bar: 100 µm).

**Figure 2 molecules-25-04950-f002:**
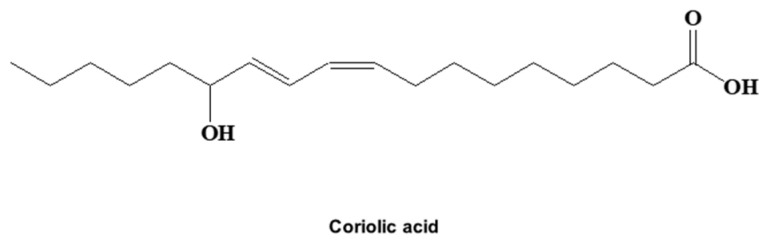
Molecular structure of coriolic acid, the cancer stem cell (CSC) inhibitor isolated from *S. herbacea* L.

**Figure 3 molecules-25-04950-f003:**
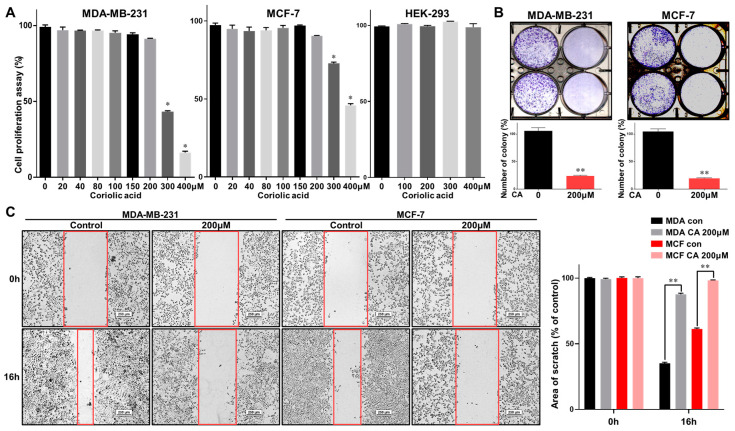
The effects of coriolic acid on cancer cell viability and mammosphere-forming efficiency. (**A**) MDA-MB-231, MCF-7, and HEK-293 cells were cultured with coriolic acid for 1 day. The cytotoxicity of coriolic acid was tested using the MTS assay. (**B**) Coriolic acid inhibits the colony formation of breast cancer cells incubated and treated with coriolic acid. (**C**) The effects of coriolic acid on the migration of breast cancer cell lines. Migrations with/without coriolic acid were imaged at 0 and 16 h (scale bar: 100 µm). The percent of inhibition in cell migration was expressed using untreated well at 100%. (**D**) Coriolic acid suppresses mammosphere-forming capacity. To make mammospheres, 0.5 × 10^4^ MDA-MB-231 cells and 4 × 10^4^ MCF-7 cells per well were plated in 6-well ultralow-attachment 6-well plates. The mammospheres were then cultured with coriolic acid. Representative mammospheres in the photos were obtained by inverted light microscopy (scale bar: 250 µm). The MFE was determined. Representative data were collected. The data from triplicate experiments are represented as the mean ± SD; * *p* < 0.005, ** *p* < 0.01.

**Figure 4 molecules-25-04950-f004:**
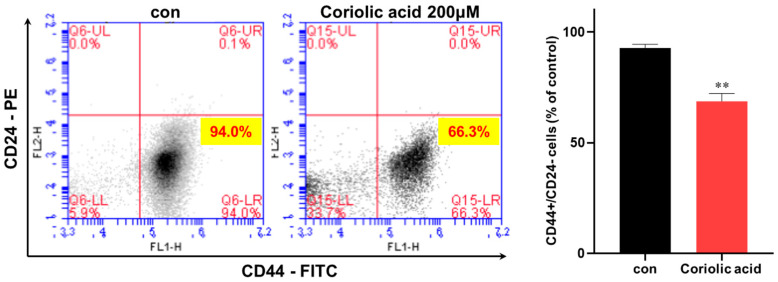
The effects of coriolic acid on the CD44^high^/CD24^low^-expressing cells within an MDA-MB-231 cell population. The CD44^high^/CD24^low^ subpopulation treated with/without coriolic acid (200 µM) for 1 day was tested by flow cytometry. For this analysis, 2 × 10^4^ cells were utilized. The red cross was based on the binding of an antibody without coriolic acid. The data are represented as the mean ± SD; ** *p* < 0.01.

**Figure 5 molecules-25-04950-f005:**
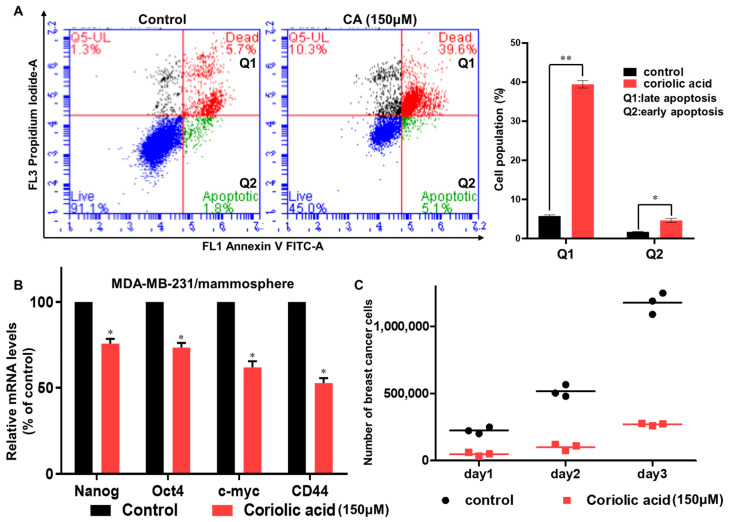
The effects of coriolic acid on apoptosis, cancer stem cell markers and mammosphere growth. (**A**) Coriolic acid increased apoptosis in BCSCs. Mammospheres were plated and cultured with coriolic acid. We cultured mammospheres in ultra-low attachment 6 well plate for 5 days and then treated 150 µM of coriolic acid for 2 days. After collecting the cells, the cells were trypsinized to be single cells and washed with 1X phosphate-buffered saline (PBS). The single cells (1 × 10^5^) were counted and suspended with 100 µL of 1X Annexin V binding buffer. We added 5 µL of FITC Annexin V solution and 5 µL of Propidium Iodide staining solution to each samples. Then, those are incubated for 15 min on room temperature. After washing with 1X Annexin V binding buffer, the cell pellet was analyzed using an Accuri C6 flow cytometer (BD, San Jose, CA, USA). Apoptosis was analyzed by annexin V/propidium oxide (PI) staining after treatment. ** *p* < 0.01, * *p* < 0.05 vs. the DMSO-treated control. (**B**) Transcriptional levels of CSC markers, including the Nanog, Oct4, c-myc, and CD44 genes, were determined in mammospheres with/without coriolic acid treatment using CSC marker-specific primers and real-time PCR (Appendix A). β-Actin was used as an internal control. The data shown represent the mean ± SD of three independent experiments. * *p* < 0.05 vs. the DMSO-treated control. (**C**) Mammosphere growth is reduced by coriolic acid. Mammospheres treated with/without coriolic acid were separated into a single cell, and the single cells were plated in equal numbers in 6 cm diameter dishes. One, two, and three days later, the cells were counted.

**Figure 6 molecules-25-04950-f006:**
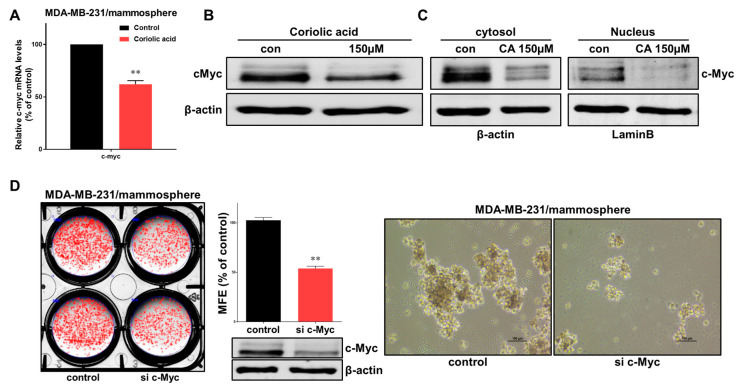
Coriolic acid regulates c-Myc mRNA and c-Myc protein levels. (**A**) Transcripts level of c-myc genes in CSCs was measured in mammospheres with/without coriolic acid treatment using c-myc primers and real-time PCR (Appendix A). β-Actin was used as an internal control. (**B**) The total protein level of c-Myc was assayed in MDA-MB-231 cell mammospheres after treatment with coriolic acid (0 or 150 µM) for 1 day using immunoblotting. Total lysates were used to analyze immunoblots with an anti-c-Myc antibody. β-Actin was used as an internal control. (**C**) After treatment with coriolic acid for 48 h, the protein levels of c-Myc in the cytosolic and nuclear fractions were analyzed in mammospheres using western blotting. Nuclear and cytosolic proteins were separated on SDS-PAGE gels, followed by immunoblotting with anti-c-Myc, anti-β-actin, and anti-lamin B antibodies. (**D**) Effects of the c-Myc protein on mammosphere formation using siRNA targeting c-Myc. C-Myc-downregulated MDA-MB-231 cells were cultured for 8 days. Images were obtained by inverted light microscopy at 10× magnification. Data are presented as the mean ± SD of three independent experiments. ** *p* < 0.01 vs. the DMSO-treated control group.

**Figure 7 molecules-25-04950-f007:**
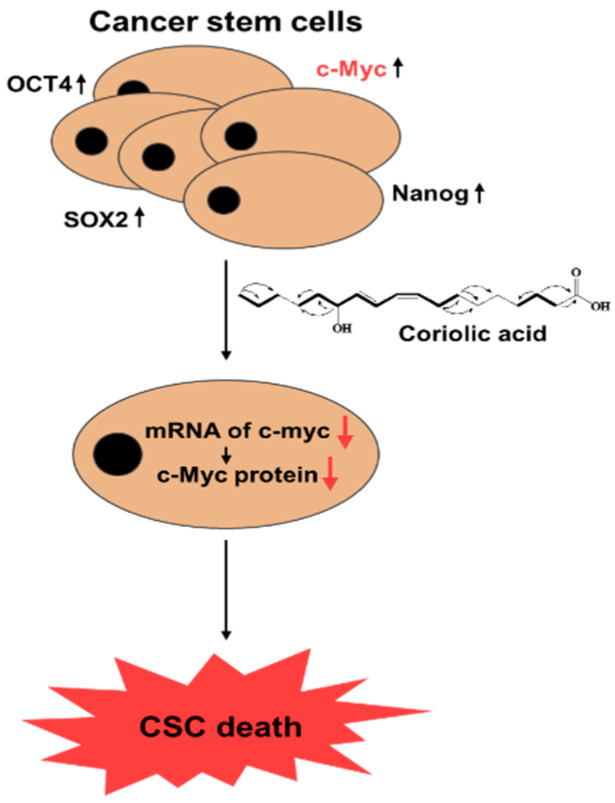
The proposed model for coriolic acid-induced CSC death.

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
