# Peer review of "Coriolic Acid (13-(S)-Hydroxy-9Z, 11E-octadecadienoic Acid) from Glasswort (Salicornia herbacea L.) Suppresses Breast Cancer Stem Cell through the Regulation of c-Myc"

_molecules, 2020, doi:10.3390/molecules25214950_

Round 1

Reviewer 1 Report

This manuscript describes the bioassay-guided isolation of the natural compound coriolic acid, a lineolic acid derivative, from the halophytic coastal plant Salicornia herbacea L. and its identification by NMR spectroscopy and electrospray ionization mass spectrometry. Subsequently the inhibitory effects of the compound on formation of mammosphere, breast cancer stem cell formation, the CD44high/CD24low subpopulation and the transcriptional levels of CSC-related genes were determined using inter alia cell viability assays and breast cancer cell lines.

The topic of the manuscript is highly relevant to the field of natural product analysis. The work is innovative, relevant and very well conducted. The manuscript is clearly structured with the conclusion supported by the data.

However, before a publication in Molecules can be considered, this manuscript would require revision considering the two comments listed below.

Comment 1: In the material and methods section on Page 9, Line 23 the vendor, column type, particle size and pore size of the chromatographic material should be provided.

Comment 2: The manuscript would benefit from minor correction of English language e.g.:

Page 2, Line 7: Replace “plant” with “salt tolerant plant”.

Page 2, Line 24: Replace “analyzed to be” with “identified as”.

Page 6, Line 10: Replace “dishes” with “diameter dishes”.

References: Please italicize species names.

Reference 15: Replace “Salicornia Herbacea” with “Salicornia herbacea”.

Author Response

We submitted notes to reviewer 1 comments.

Reviewer 2 Report

The authors describe bioactivity of coriolic acid, which they identified as an active compound from Salicornia herbacea L. acting by anticancer mechanism against breast cancer stem cells (BCSCs). The study and the results gained are interesting, but several issues should be solved before a possible publication of this article.

Results

2.2 – very poorly described results of the cell viability assay, must be improved, further it is totally unclear why HEK 293 – kidney cells were used for this assay…?

Do you think that such high (hundreds of micromolar concentrations) have biological relevance?

pH of the media should be definitively measured at the highest concentrations of coriolic acid added to exclude the effect made only by a significant pH decrease than the compound itself. This is a must !

Figure 3 – poor resolution, too small, almost unreadable even after zooming, must be improved, you should dissect the image into more images so that it is readable, wound healing assay – scale bars are missing, on the right part of these wound healing images, there is some dark spot (right part, bottom of the image) which is the same at each image – weird – what is it? Surface area of the wounds should be calculated and plotted, also statistical analysis should be done (from how many samples and how many measurements it was done?

2.3. an initial sentence stating what was expected from the experiment logically connecting the individual chapters and creating flow, is missing, add error to the percentages stated in this chapter, otherwise the results have no meaning

Figure 4 – too small, almost unredable, too much white space around, why so big concentration was used? The impact on surface protein expression should be present at much lower concentrations than those that decrease cell viability/are killing the cells, also shorter time than 1 day should be examined together with this time point discussed

2.4 again the results without stating the errors are meaningless, the results are described very poorly/vaguely, also time points used are missing, as well as concentrations used in this measurement are not discussed

Figure 5 – too small, almost unreadable, too much white space around

Figure 6 – again, too small, unreadable, the fractionation, which is described in the caption is not described in the methods? Why? Scale bars missing in the microscopy images

Figure 7 – not very informative, try to improve, elaborate on the model more

Discussion:

Is quite poor, should be extended and the authors should discuss the results more in the view of the mechanism of action and cellular events (with the literature)

Methods:

Why did they culture the cells with antibiotics? It can significantly influence the results and it also does not speak about good laboratory practice and sterile work with cell culture.

Chapter 4.5 is called “Breast Cancer Cell and Mammosphere Cultures“ but you describe also HEK 293, these are not breast cells and not cancer cells ! Why were HEK 293 cells used in this study?

4.6 – what were the controls, how many biological and technical replicates? Describe the measurement more / better in detail, not only “according to manufacturer”, also in what was the coriolic acid dissolved? You do not even write here the method, which was used, one can find it only in the results that it was MTS

4.9 describe better the experiment in more details so that it is reproducible, also write the concentrations of the antibodies used

4.10 describe better the experiment in more details so that it is reproducible

4.11 describe better the experiment in more details so that it is reproducible, also write the concentrations of the antibodies used, also what was the SDS-PAGE setup (tris-glycine or tris-tricine,…), describe all the methods so that other researcher can reproduce them, all the method are described very vaguely

4.12 describe better the experiment in more details so that it is reproducible

Conclusion:

Very, very poor, you have to elaborate on this part a lot

Author Response

We submitted a note of reviewer 2 comments.

Round 2

Reviewer 2 Report

The authors answered most of the questions and concerns, the article has significantly improved and I recommend it for publication.